# Key Parameters on the Antibacterial Activity of Silver Camphor Complexes

**DOI:** 10.3390/antibiotics10020135

**Published:** 2021-01-30

**Authors:** Joana P. Costa, Sílvia A. Sousa, Adelino M. Galvão, J. Miguel Mata, Jorge H. Leitão, M. Fernanda N. N. Carvalho

**Affiliations:** 1Centro de Química Estrutural and Departamento de Engenharia Química, Instituto Superior Técnico, Universidade de Lisboa, Av. Rovisco Pais, 1049-001 Lisboa, Portugal; joanavcosta@tecnico.ulisboa.pt (J.P.C.); adelino@tecnico.ulisboa.pt (A.M.G.); josemiguelmata4@tecnico.ulisboa.pt (J.M.M.); 2IBB-Institute for Bioengineering and Biosciences, Department of Bioengineering, Instituto Superior Técnico, Universidade de Lisboa, Av. Rovisco Pais, 1049-001 Lisboa, Portugal; sousasilvia@tecnico.ulisboa.pt

**Keywords:** silver complexes, camphor imine ligands, antibacterial activity, complex design, redox properties

## Abstract

Nine new complexes with camphor imine or camphor sulfonimine ligands were synthesized and analytically and spectroscopically characterized, aiming to identify the key parameters that drive the antibacterial activity of the complexes with metal cores and imine substituents with distinct electronic and steric characteristics. The antimicrobial activity of all complexes was evaluated by determining their minimum inhibitory concentrations (MIC) against the Gram-negative *Escherichia coli* ATCC25922, *Pseudomonas aeruginosa* 477, and *Burkholderia contaminans* IST408, and the Gram-positive *Staphylococcus aureus* Newman. Camphor imine complexes based on the hydroxyl silver center ({Ag(OH)}) typically perform better than those based on the nitrate silver center ({Ag(NO_3_)}), while ligands prone to establish hydrogen bonding facilitate interactions with the bacterial cell surface structures. A different trend is observed for the silver camphor sulfonimine complexes that are almost non-sensitive to the nature of the metal cores {Ag(OH)} or {Ag(NO_3_)} and display low sensitivity to the Y substituent. The antibacterial activities of the Ag(I) camphor sulfonimine complexes are higher than those of the camphor imine analogues. All the complexes display higher activity towards Gram-negative strains than towards the Gram-positive strain.

## 1. Introduction

New antimicrobials are urgently needed to overcome the increasing resistance of bacteria to existing antibiotics. In the European Union (EU), bacterial infections pose a considerable burden on health care systems, accounting for ca. 33,000 deaths in 2015 [1]. Multi-resistant *Klebsiella pneumoniae*, *Enterobacter* sp., *Pseudomonas aeruginosa*, or *Staphylococcus aureus*, members of the ESKAPE group of pathogens, deserve special attention since they are prevalent in hospitalized patients with depressed immune systems and some strains are resistant to the third generation β-lactam antibiotics, as well as carbapenems [2].

In our days, threats to human health include not only the emergence of multi-resistance among common microbial human pathogens, but of also emerging new pathogens, as is the case of SARS-COV2. These health threats create a worldwide challenge and pressure to the discovery and development of new antimicrobials, a domain that has not been intensively pursued by the pharma industry in the last decades [3]. 

Thus far, the library of antimicrobials relies mostly on organic compounds. However, coordination compounds may be eligible alternatives with beneficial properties [4] due to the characteristics of the metal. In addition, at the complexes the ligands may be activated towards a reactivity distinct from that they present as free organic entities or be released leaving vacant coordination positions at the metal site that may enable interactions with bacterial cell components. The redox properties of coordination compounds are among the parameters that distinguish them from the organic compounds and may play a relevant role in their mechanisms of antibacterial activity. Electron transfer processes are of utmost relevance in biological processes. Compounds that interfere with biological electron transfer processes may trigger the formation of reactive organic species (ROS), leading, for example, to the inhibition of the bacterial cell respiratory chain. The electronic and steric characteristics of coordination compounds may also switch catalytic redox processes (on/off) reducing bacteria growth, e.g., through ROS generation [4].

The antibacterial properties of silver were recognized long ago by Hippocrates (the father of medicine). During the last century, silver sulfadiazine was used to control infections on wounds and burns [5], despite some concerns raised due to possible side effects [6]. Over the two last decades, research focused on the search for potential antimicrobial properties of silver coordination compounds raised considerably. Several families of silver-based coordination compounds have been shown to display relevant antibacterial activity with acceptable toxicity [7,8,9,10,11].

Within our work, we found that several camphor imine complexes have high antibacterial activity, often combined with antifungal activity [12,13]. Such activities can be suitably engineered through the design of the camphor imine ligands and the inner sphere of the metal through co-ligands to optimize the activity of the complexes towards specific targets.

In the present work, several sets of camphor imine ([Ag(NO_3_)(XC_10_H_14_NY)] (X= O or N, Series **a**), [Ag(OC_10_H_14_NY)_2_((μ-O)] (Series **b**), [Ag(OH)(OC_10_H_14_NY)_2_] (Series **c**)), and camphor sulfonimine complexes ([Ag(NO_3_)(O_2_SNC_10_H_14_NY)_2_] (Series **d**), [Ag(OH)(O_2_SNC_10_H_14_NY)_2_] (Series **e**), and [Ag(OH)(O_2_SNC_10_H_14_NY)] (Series **f**)) were synthesized, analytically and spectroscopically characterized, and their antibacterial activities determined, in order to get insights into the effect of structural changes at the camphor ligand or inner sphere on the antibacterial properties of the complexes.

## 2. Results and Discussion

### 2.1. Synthesis

Six sets of silver complexes were synthesized based on camphor-type ligands, as shown in Figure 1, aiming at enhancing the antibacterial properties of Ag(I) camphor imine complexes [Ag(NO_3_)L_2_] formerly reported [12,13].

[Ag(NO_3_)L] (Series **a**) differ on the metal to ligand ratio (1:1) from the previous nitrate complexes [Ag(NO_3_)L_2_] (1:2) and, consequently, on the structural arrangement. It was not possible to obtain suitable crystals from [Ag(NO_3_)(OC_10_H_14_NC_6_H_4_NH_2_−4)] (**1a**) for X-ray diffraction analysis. However, [Ag(NC_10_H_14_NC_6_H_4_)]NO_3_ (**6a**) was structurally characterized by X-ray diffraction, showing that it arranges as a coordination polymer, as shown in Figure 2. 

The unit cell of complex **6a** is formed by a silver ion (Ag(I)) coordinated to the nitrogen atom of one of the imine nitrogen atoms of the ligand. This unit bridges the Ag(I) site of a neighbor unit through the amine nitrogen atom of the ligand, forming a one-dimensional network (coordination polymer, Figure 2c) (N-Ag-N = 168.6(2) deg). A parallel 1D alignment of the nitrate ion (NO_3_^−^) equilibrates the positive charge of the silver cationic units, as shown in Figure 2c. Only one of the oxygen atoms in the nitrate anion is at nearly bonding distance from the silver atom (Ag–O51, 252.5(7) pm), occupying an orthogonal position with respect to the imine nitrogen atom (95.9(5) and 94.8(5) deg), generating a T-shaped ML_3_ metal center, as shown in Figure 2a. 

Analytical data shows that **6a** and **1a** have in common the 1:1 metal to ligand ratio and conceivably the same polymeric arrangement. However, they differ in the ionic (**6a**, λ_M_ = 142 ohm^−1^ cm^2^ mole^−1^) and neutral (**1a**, λ_M_ = 64 ohm^−1^ cm^2^ mole^−1^) character, as verified through measurement of conductivity in acetonitrile solutions (1 × 10^−3^ M) [14]. The ionic character of **6a** was confirmed by X-ray analysis. In order to get further insights into the structural differences of **1a** and **6a**, calculations using density functional theory (DFT) were undertaken for **1a** based on an oligomer as a model compound for the coordination polymer. The oligomer was built by adding “mer” units incrementally making the chain grow. The four silver camphor units, as shown in Figure 3, can explicitly show the structural trends of the polymer as an illustrative example. Further computational details are given in Section 3.

Results showed that at **1a,** the camphor ligand bridges two {Ag(NO_3_)} units—one through the imine nitrogen atom of the hydrazine moiety (NNH_2_), and the other through the oxygen atom (C = O) of the camphor skeleton. Additionally, the amine substituent (Y = NH_2_) bridges one of the oxygen atoms of the mono coordinated nitrate (NO_3_^−^) co-ligand, as shown in Figure 3, stabilizing the polymeric neutral character of the complex. These computational calculations extend the number of distinct structural arrangements found for polymeric Ag(I) camphor imine complexes, highlighting the structural versatility of the Ag(I) camphor imine system. Such versatility was also found for mononuclear complexes [Ag(NO_3_)(OC_10_H_14_NY)_2_] (Y = 2−5, Figure 1), that depending on the characteristics of the imine substituent (Y group) may display distorted octahedral to trigonal prismatic or even linear geometries [13].

In order to ascertain whether changing the co-ligand also changes the structure and properties of the complexes, nitrate was replaced by bridging oxygen or hydroxy ligands affording [{AgL}_2_(μ-O)] (Series **b**) and [Ag(OH)L] (Series **c**), respectively. Series **b** displays a dimer character with two Ag^I^L units bridged by oxygen, while Series **c** keeps the 1:1 metal to ligand ratio according to a polymeric arrangement as found for **1a**. The general formula found for silver hydroxy complexes follow those found for the nitrate complexes except that no complexes that fit the 1:2 metal to ligand ratio were obtained. 

To check whether steric changes in the ligand affect the characteristics of the complexes, three sets of Ag(I) camphor sulphonyl imine complexes were synthesized.

Camphor sulphonyl imines (O_2_SNC_10_H_13_NY), as shown in Figure 1, differ from camphor imines (OC_10_H_14_NY) in an extra five members ring annulated to the camphor skeleton and presence of the sulphonyl imine group (NSO_2_) while keeping the imine substituent (Y, at position 3, Figure 1). For comparison purposes, a new camphor sulfonimine ligand (SO_2_NC_10_H_13_NC_6_H_4_CH_3_−4) was synthesized (see Section 3) and structurally characterized by X-ray diffraction analysis, as shown in Figure 4. 

Using camphor sulphonyl imines as ligands, Ag(I) nitrate complexes [Ag(NO_3_)(SO_2_NC_10_H_13_NY)_2_] (Series **d**), bridging oxygen [{Ag(SO_2_NC_10_H_13_NY)_2_}_2_(μ-O)] (Series **e**), and hydroxy [Ag(OH)(SO_2_NC_10_H_13_NY)] (Series **f**) co-ligands were synthesized. 

All the new complexes of the six families (Series **a** to Series **f**) were analytical and spectroscopically characterized by NMR (^1^H, ^13^C, DEPT, HMBC, HSQC) and FTIR (see Section 3). 

### 2.2. Antibacterial Activity

The antibacterial activity of all complexes was evaluated against Gram-negative *E. coli* ATCC25922, *P. aeruginosa* 477, *B. contaminans* IST408, and Gram-positive *S. aureus* Newman through determination of the minimum inhibitory concentration (MIC). 

Results show that all complexes are active against the bacterial strains under study. The activities against Gram-negative are considerably higher than against the Gram-positive bacterial strain, as shown in Table 1. 

Complex **1a** is particularly efficient against the Gram-negative strains tested (MIC, 22−27 μg/mL), displaying a moderate activity towards the Gram-positive *S. aureus* strain Newman (MIC, 118 μg/mL). In contrast, complex **6a** (ionic) that structurally resembles **1a** (neutral) in the polymeric character displays considerably lower antibacterial activity than **1a**, as shown in Table 1. Such an observation corroborates the relevance of the neutral versus ionic character of the Ag(I) camphor imine complexes on their antimicrobial activity, as previously reported for the mononuclear complexes [Ag(NO_3_)L_2_] or [AgL_2_]NO_3_ [12]. Parameters such as polarity, size, and lipophilicity are important factors for passive diffusion, affecting the partitioning and/or diffusion of a molecule into and across the membrane. In addition, highly polar groups also significantly decrease the permeability of parent compounds by orders of magnitude [15]. Since no specific transporters are known for the studied compounds, we assume that their entry into the bacterial cell mainly occurs by diffusion processes to which the basic character of the amine group at the camphor ligand (Y = NH_2_, **1a**) and its ability to undergo hydrogen bonding in addition to the neutral character of the complex are the parameters that enhance the antibacterial performance [16]. Therefore, the differences observed in the antimicrobial activity of **1a** and **6a** can be attributed to the lower ability of **6a** to diffuse across bacterial membranes due to its ionic character and the characteristics of the Y group.

Another parameter that, according to the previous studies, affects the biological activity of the Ag(I) camphor complexes is the co-ligand [17,18]. Replacement of nitrate by hydroxide at the inner sphere of the complexes was found to switch non-active [Ag(NO_3_)L] into active [Ag(OH)L] complexes against *Candida albicans* [14]. 

The herein results further corroborate the relevance of the co-ligands into the biological activity of the camphor imine complexes (Series **a**, **b**, **c**) by showing that complexes [Ag(OH)(OC_10_H_14_NY)] (Series **c**) that structurally resemble [Ag(NO_3_)(OC_10_H_14_NC_6_H_4_NH_2_−4)] (**1a**), in the neutral coordination polymer character, display lower MIC values, as shown in Table 1, than **1a** and much lower than [{Ag(OC_10_H_14_NY)}_2_(μ-O)] (Series **b**) [19]. Thus, replacement of nitrate by hydroxide enhances the antimicrobial properties of the complexes. Complex [Ag(OH)(OC_10_H_14_NOH)] (**2c**) is specially active towards Gram-negative (3.4–7.2 μg/mL) and Gram-positive (9.2 μg/mL) bacterial strains, a fact that is attributed to hydrogen interactions with bacterial surface structures established through the co-ligand (OH) and/or the camphor imine substituent (Y = OH). On the contrary, the dimer character of complexes [{Ag(OC_10_H_14_NY)}_2_(μ-O)] (Series **b**) disfavors the antibacterial activity to MIC values in the range of the former reported for [Ag(NO_3_)L_2_] [12], although slightly higher than that of the ionic **6a**, as shown in Table 1. 

To compare the antibacterial activity of the camphor imine with the camphor sulfonimine Ag(I) complexes, the MIC values for Series **d**, **e**, and **f** were evaluated, as shown in Table 1. Data shows that all camphor sulfonimine complexes (except **9e**) have high antibacterial activities which are comparable or even higher than that of camphor imine complexes. For example, complex **8d** (camphor sulfonimine ligand) and **1a** (camphor imine ligand), which have in common the metal center {Ag(NO_3_)} and the camphor substituent (Y = NH_2_) and differ on the mononuclear/polymeric character, display similar MIC values for *E. coli* (22 μg/mL, **1a**, 27 μg/mL, **8d**) while **8d** performs slightly better for *P. aeruginosa*, *B. contaminans*, and *S. aureus*, as shown in Table 1. In what concerns the effect of the co-ligand on the MIC values of the complexes, it comes out that the camphor sulfonimine complexes are less sensitive than camphor imine complexes to changes in the silver core. Complexes of Series **d** and **f** display values in the same range for the four bacterial strains, although their metal sites are of types [Ag(NO_3_)L_2_] and [Ag(OH)L_2_], respectively, as shown in Table 1.

For a better rationalization of all the data it would be necessary to get the full set of complexes (with the same Y substituents) for the six families of compounds. However, undesired reactions between the metal salts and the ligand precursors prompted redox or other processes leading to the formation of Ag^0^ or silver oxides, instead of the coordination compounds. 

A rationalization of the above MIC values, as shown in Table 1, shows that the Ag(I) complexes based on camphor imine or camphor sulfonimine behave differently in what concerns the {Ag(OH)} or {Ag(NO_3_)} metal cores and Y substituents. The activity of the camphor sulfonimine complexes (Series **d**, **f**, **e**) is basically independent of the co-ligand at silver site and slightly dependent of Y. The camphor sulfonimine ligand somehow buffers the process. A different trend is detected for the camphor imine silver complexes (Series **a**, **b**, **c**) that reveal activities that depend on (i) the ionic or neutral character, (ii) the characteristics of the Y substituent at the camphor imine group, and (iii) the co-ligand at metal site.

The MIC values obtained for the complexes and the precursor silver salts, as shown in Table 1, show that except **6a** (ionic) and **3b**, **5b**, **9e** (oxygen bridged), the complexes have higher activities than their precursors. The ligands by themselves are inactive.

### 2.3. Redox Properties

Biological processes commonly involve electron transfer and thus redox processes [20,21]. In order to evaluate whether a relationship exists between the redox characteristics and the antimicrobial properties of the complexes, their electrochemical properties were studied by cyclic voltammetry, as shown in Table 1. The results show that {Ag(NO_3_)} complexes (**1a**−**6a** and **8d**−**9d**) reduce at potentials higher than {Ag(OH)} (**2c**, **4c**, **7c**, **10f**, and **11f**) complexes, as shown in Table 1, and both families of complexes reduce at lower potentials then their precursors Ag(NO_3_) ($E_{1/2}^{red}=$ 0.18 V) or Ag(OAc) ($E_{1/2}^{red}=$ −0.043 V), evidencing that the complexes are more electron rich than their precursors. In all cases, adsorption waves are observed in the reverse scan upon reduction, as shown in Figure 5a, that reveal silver metal formation (Ag^0^). The adsorption waves are less pronounced in the case of Ag(OH) than Ag(NO_3_)-based complexes, conceivably due to the fact that the {Ag(OH)}-based complexes are considerably more difficult to reduce than {Ag(NO_3_)}-based complexes. A peculiar feature of the silver hydroxide complexes is that Ag(I)→Ag(0) reduction occurs at potentials in the range of the ligands ($E_{1/2}^{red}=$ −1.46 V, **8** [22]; $E_{1/2}^{red}=$ −1.16V, **9**; $E_{1/2}^{red}=$ −1.25 V, **10**; $E_{p}^{red}$ = −1.37 V, **11**).

Representative cyclic voltammograms, as shown in Figure 5, highlighting the Ag(I)→Ag(0) reduction wave are displayed for complexes **5a**, **7c**, **9d**, and **10f**. Dotted lines refer to the cyclic voltammogram of the free ligands, as shown in Figure 5b,d, evidencing proximity to metal reduction. 

The rationalization of the antibacterial and electrochemical data shows that complexes which reduce at low potentials display low MIC values, except **9e** that displays low potential and low activity. Additionally, complexes able to bond with hydrogen (**1a**, **2c**, **8d**) can overcome the unfavorable effect of the high Ag(I) reduction potential and display high antibacterial activity at least for some bacteria strains, as shown in Figure 6. 

We are aware that generalization must be careful. Nevertheless, low Ag(I)→Ag(0) potentials are consistent with improved resistance to reduction by electron donors in the cell (e.g., pyocyanin) [23], becoming a relevant parameter to the stability of the complexes in the biological medium and enhancement of their antibacterial activity. 

## 3. Materials and Methods

### 3.1. General

The new complexes were synthesized under nitrogen using Schlenk and vacuum techniques unless stated otherwise. Camphor ligands (OC_10_H_14_NY: Y = NH_2_, OH, C_6_H_5_, C_6_H_4_NH_2_−4, C_6_H_4_CH_3_−4, and C_6_H_4_OH−3) were prepared according to reported procedures [24]. Silver salts, camphor, camphorsulfonic acid, the amines, and hydrazine were purchased from Sigma Aldrich. Acetonitrile (PA grade) was purchased from Carlo Erba and was purified by conventional techniques [25,26] and distilled before use. The FTIR spectra were obtained from KBr pellets using a JASCO FT/IR 4100 spectrometer. The NMR spectra (^1^H, ^13^C, DEPT, HSQC, and HMBC) were obtained from CD_3_CN, CD_2_Cl_2_, CDCl_3_, or DMSO solutions using Bruker Avance II+ (300 or 400 MHz) spectrometers. The NMR chemical shifts are referred to TMS (δ = 0 ppm).

### 3.2. Synthesis 

#### 3.2.1. Ligands 

SO_2_NC_10_H_13_NC_6_H_4_NH_2_ (**10**)—Oxoimine (454 mg; 2 mmol) was dissolved in ethanol (7 mL), and the solution was acidified with CH_3_COOH (0.2 mL) and stirred for 20 minutes. Then, benzene−1,4-diamine (216 mg; 2 mmol) was added and the flask was saturated with N_2_. An orange precipitate formed upon overnight stirring at 40 °C that was filtered off solution affording the compound. Yield 77%. FTIR (KBr, cm^−1^): 3484 (NH_2_); 3384 (NH_2_); 1641 (C = N); 1317 (SO_2_); 1161 (SO_2_). ^1^H NMR (CDCl_3_, δ_ppm_): 7.03 (d, *J* = 7.6 Hz, 2H); 6.70 (d, *J* = 7.7 Hz, 2H); 3.90 (s, 2H); 3.36, 3.17 (2d, *J* = 13.5 Hz, 2H); 3.15 (d, *J* = 3.8 Hz, 1H); 2.33−2.17 (m, 2H); 2.04−1.97 (m, 1H); 1.87−1.79 (m, 1H); 1.10 (s, 3H); 0.90 (s, 3H). 

^13^C NMR (CDCl*_3_*, δ_ppm_): 186.0 (C2); 161.9 (C3); 146.7 (C_ipso_); 139.4 (C_ipso_); 125.3 (C_Ph_); 115.2 (C_Ph_); 62.6 (C1); 52.0 (C4); 50.0 (C8); 47.4 (C7); 29.0, 23.9 (C5, C6); 20.0, 18.7 (C9, C10).

SO_2_NC_10_H_13_NC_6_H_4_CH_3_ (**11**)—3-oxo-camphorsulfonimide (454 mg; 2 mmol) was dissolved in ethanol (7 mL) and the solution was acidified with CH_3_COOH (0.2 mL). Upon stirring for 20 minutes, *p*-toluidine was added (268 mg; 2 mmol) and the mixture stirred at T = 40 °C overnight. A yellow precipitate formed that was filtered off affording the yellow compound. Yield 78%. FTIR (KBr, cm^−1^): 1658; 1633 (C = N); 1337 (SO_2asym_); 1161 (SO_2sym_). ^1^H NMR (CDCl_3_, δ_ppm_): 7.22 (d, *J* = 7.0 Hz, 2H); 6.91 (d, *J* = 7.1 Hz, 2H); 3.38, 3.18 (2d, *J* = 13.1 Hz, 2H); 3.05 (d, *J* = 3.7 Hz, 1H); 2.28−2.18 (m, 2H); 2.07−1.98 (m, 1H); 1.85−1.76 (m, 1H); 1.09 (s, 3H); 0.93 (s, 3H); 0.77 (s, 3H). ^13^C NMR (CDCl_3_, δ_ppm_): 185.5 (C2); 166.2 (C3); 146.4, 136.9 (C_ipso_); 129.9, 121.2 (C_Ph_); 62.8 (C1); 51.8 (C4); 50.1 (C8); 46.9 (C7); 28.7, 24.2 (C5, C6); 20.1, 18.6 (C9, C10). 

#### 3.2.2. Complexes 

Due to sensitivity of Ag(I) solutions to light, the flasks with the reaction mixtures were covered with aluminum foil. Complexes **2a**, **3a**, **4a**, **5a**, **3b**, **5b**, and **8d** were prepared following the procedures previously reported [12,13,14].

[Ag(NO_3_)(OC_10_H_14_NNH_2_)] (**1a**)—The camphor imine ligand OC_10_H_14_NNH_2_ (**1**, 0.090g; 0.50 mmol) was dissolved in acetonitrile (7 mL). Then, silver nitrate (0.085 g; 0.50 mmol) was added and the mixture was stirred for ca. 8 hours. A light grey suspension (silver particles) was obtained that was filtrated off solution. The transparent solution was then evaporated under vacuum until formation of an oil to which Et_2_O (3 mL) was added. A white precipitate formed upon solvent evaporation. Yield 50%. Elem. Anal. (%) for AgC_10_H_16_N_3_O_4_: Found: C, 34.0; N, 11.7; H, 4.8. Calc.: C, 34.3; N, 12.0; H, 4.6. FTIR (KBr, cm^−1^): 3414, 3294 (NH); 1707 (C = O); 1577 (C = N); 1384 (NO_3_). ^1^H NMR (CD_2_Cl_2_, δ_ppm_): 7.40 (sl, 2H); 3.20 (d, *J* = 3.7 Hz, 1H); 2.04−1.95 (m, 1H); 1.79−1.70 (m, 1H); 1.60−1.43 (m, 2H); 0.95 (s, 6H); 0.77 (s, 3H). ^13^C NMR (CD_2_Cl_2_, δ_ppm_): 204.5 (C2); 155.0 (C3); 58.2 (C1); 46.8 (C7); 45.7 (C4); 31.2, 23.0 (C5, C6); 20.5, 17.2 (C9, C10); 8.7 (C8). 

[Ag(NC_10_H_14_NC_6_H_4_)]NO_3_ (**6a**)—AgNO_3_ (170 mg; 1.0 mmol) and the ligand (**6**, C_16_H_18_N_2_, 238 mg; 1.0 mmol) were mixed in a Schlenk and stirred under vacuum for 15 minutes. Dried acetonitrile (10 mL) was then added, and the mixture was stirred over-night at room temperature. The suspension was filtered to remove silver particles. From the solution the complex precipitated as an off-white solid upon partial solvent evaporation of the solvent. The solid was filtered off from the colorless solution and by further evaporation of the solvent followed by a few days in the freezer, another crop of the complex was obtained that was filtered off solution. Yield 77%. Elem. Anal. (%) for AgC_16_H_18_N_3_O_3_·⅟_4_Et_2_O: Found: C, 41.0; N, 11.4; H, 5.2. Calc.: C, 41.2; N, 11.1; H, 5.5. FTIR (KBr, cm^−1^): 1517 (C = N); 1360 (NO_3_). ^1^H NMR (CD_3_CN, δ_ppm_): 7.98−7.96 (m, 2H), 7.68−7.66 (m, 2H), 3.03 (d, *J* = 4,4Hz, 1H), 2.37−2.27 (m, 1H), 2.14−2.04 (m, 1H), 1.98−1.92 (m, 2H), 1.38 (s, 3H), 1.12 (s, 3H), 0.58 (s, 3H). ^13^C NMR (CD_3_CN, δ_ppm_): 167.0 (C2); 165.5 (C3); 142.6, 141.6 (C_ipso_); 129.7, 129.5, 129.4, 129.3 (C_Ph_); 55.1 (C1); 54.8 (C7); 54.7 (C4); 32.4, 25.1 (C5, C6); 20.4 (C9); 18.6 (C10); 10.4 (C8). 

[Ag(OH)(OC_10_H_14_NOH)]· ½EtOH (**2c**)—A solution of the ligand NC_10_H_14_N(OH) (**2**, 54 mg; 0.30 mmol) in ethanol (5 mL) was added to a solution of silver acetate (50 mg; 0.30 mmol) in water (5 mL). The mixture was stirred for 2 hours at room temperature. Then, the solution was filtered to separate a slight suspension and the solution evaporated to dryness affording an off-white compound. Yield 51%. Elemental analysis for AgC_10_H_16_NO_3_·½EtOH, Exp.: C, 39.9; N, 3.9; H, 5.4. Calc.: C, 40.1; N, 4.3; H, 5.8. IR (KBr, cm^−1^): 3444 (OH), 1743 (CO), 1643 (CN). ^1^H NMR (DMSO, δ_ppm_): 3.08 (d, *J* = 3.7 Hz, 1H); 1.98−1.88 (m, 1H); 1.80 (s, 1H); 1.79−1.69 (m, 1H); 1.41−1.26 (m, 2H); 0.92 (s, 3H); 0.90 (s, 3H); 0.77 (s, 3H). ^13^C NMR (DMSO, δ_ppm_): 203.8 (C2); 96.6 (C3); 57.9 (C1); 45.9 (C4); 44.3 (C7); 30.1, 23.4 (C5, C6); 20.1 17.3 (C9, C10); 8.8 (C8). The chemical shifts of EtOH are omitted for clarity.

[Ag(OH)(OC_10_H_14_N(C_6_H_4_OH−3)] (**4c**)—A solution OC_10_H_14_NC_6_H_4_OH−3 (**4**, 77 mg; 0.30 mmol) in ethanol (5 mL) was added to a solution of silver acetate (50 mg; 0.30 mmol) in water (5 mL) and the mixture was stirred for 2 hours. A slight suspension formed that was filtered off and the solution evaporated to dryness providing a dark-green compound. Yield 52%. Elemental analysis for AgC_16_H_20_NO_3_, Exp.: C, 50.5; N, 3.5; H, 5.3. Calc.: C, 50.3; N, 3.7; H, 5.3. FTIR (KBr, cm^−1^): 3434 (OH); 1748 (CO); 1662 (CN). ^1^H NMR (CD_3_CN, δ _ppm_): 7.20 (t, *J* = 7.9 Hz, 1H), 6.66 (dd, *J* = 6.3, 1.9 Hz, 1H), 6.39 (s, 1H), 6.36 (d, *J* = 2.4 Hz, 1H), 2.76 (d, *J* = 4.8 Hz, 1H); 2.19 (s, OH); 1.91−1.84 (m, 2H); 1.63−1.56 (m, 2H); 1.03 (s, 3H); 0.97 (s, 3H); 0.85 (s, 3H). ^13^C NMR (CD_3_CN, δ_ppm_): 207.5 (C2); 173.3 (C3); 158.9, 152.4 (C_ipso_); 131.1, 113.0, 112.2, 107.8 (C_Ph_); 59.0 (C1); 51.2 (C4); 45.2 (C7); 30.9, 24.9 (C5, C6); 21.2, 17.6 (C9, C10); 9.4 (C8).

[Ag(OH)(OC_10_H_14_NC_6_H_4_NH_2_−4)]. CH_3_COOH (**7c**)—A solution of the ligand (**7**, OC_10_H_14_NC_6_H_4_NH_2_−4, 76 mg; 0.30 mmol) in ethanol (5 mL) was added to a suspension of silver acetate (50 mg; 0.30 mmol) in water (5 mL) and N_2_ was fluxed through the mixture that was stirred for 5 hours at room temperature. The suspension was filtered off to remove residues and the clear solution was evaporated affording a black-bright compound. Yield 57%. Elem. Anal. for AgC_16_H_21_N_2_O_2_·CH_3_COOH Exp.: C, 49.3; N, 6.2; H, 5.4. Calc.: C, 49.0; N, 6.4; H, 5.7. FTIR (KBr, cm^−1^): 3455, 3340 (NH_2_); 1733 (CO); 1625 (CN); 1567 (COO). ^1^H NMR (CD_3_CN, δ_ppm_): 6.89 (d, *J* = 7.8 Hz, 2H); 6.67 (d, *J* = 7.8 Hz, 2H); 4.30 (s, 2H); 2.97 (d, *J* = 4.8 Hz, 1H); 1.92−1.85 (m, 1H); 1.68−1.49 (m, 3H); 1.89 (s, 3H); 1.02 (s, 3H); 0.99 (s, 3H); 0.86 (s, 3H). ^13^C NMR (CD_3_CN, δ_ppm_): 208.1 (C2); 169.4 (C3); 148.4, 139.6 (C_ipso_); 125.1, 115.4 (C_Ph_); 58.7 (C1); 51.6 (C4); 45.9 (C7); 31.2, 24.6 (C5, C6); 20.9, 17.8 (C9, C10); 9.4 (C8).

[Ag(NO_3_)(SO_2_NC_10_H_13_NC_6_H_5_)_2_]·H_2_O (**9d**)—Ligand (**9**, OC_10_H_14_NC_6_H_5_, 91 mg; 0.30 mmol) in ethanol (5 mL) was added to a suspension of silver acetate (50 mg; 0.30 mmol) in water (5 mL), bubbled with N_2_ and stirred for 4 hours at room temperature. The suspension was filtered off to remove residues and the solution was evaporated affording a black-bright compound, that was filtered off affording the complex. Yield 57%. Elem. Anal. for AgC_32_H_38_N_5_O_8_S_2_·H_2_O Exp.: C, 48.6; N, 9.0; H, 4.7; S, 7.6. Calc.: C, 48.5; N, 8.8; H, 4.8; S, 8.0. IR (KBr, cm^−1^): 3434 (OH); 1667 (CN); 1637 (CN); 1384 (NO_3_); 1344 (SO_2_); 1162 (SO_2_). ^1^H NMR (DMSO, δ_ppm_): 7.46 (t, *J* = 7.4 Hz, 2H); 7.25 (t, *J* = 7.4 Hz, 1H); 6.99 (d, *J* = 8.4 Hz, 2H); 3.74, 3.52 (2d, *J* = 14 Hz, 2H); 2.88 (d, *J* = 4.8 Hz, 1H); 2.32−2.25 (m, 1H); 2.19–2.10 (m, 1H); 1.81−1.71 (m, 2H); 1.01 (s, 3H); 0.84 (s, 3H). ^13^C NMR (DMSO, δ_ppm_): 186.1 (C2); 167.8 (C3); 148.8 (C_ipso_); 129.3, 126.0, 120.1 (C_Ph_); 63.0 (C1); 51.0 (C4); 49.3 (C8); 46.3 (C7); 27.7 23.4 (C5, C6); 19.2, 17.5 (C9, C10).

[{Ag(SO_2_NC_10_H_13_NC_6_H^5^)_2_}2(µ-O)]·4H_2_O (**9e**)—A solution of the ligand (**9**, OC_10_H_14_NC_6_H_5_, 84 mg; 0.18 mmol) in ethanol (5 mL) was added to a suspension of silver acetate (30 mg; 0.18 mmol) in water (5 mL). The mixture stirred overnight at 40 °C under N_2_. The suspension was filtered off affording a dark compound. Yield 64%. Elem. Anal. for Ag_2_C_64_H_72_N_8_O_9_S_4_·4H_2_O Exp.: C, 50.4; N, 7.3; H, 4.7; S, 8.4. Calc.: C, 50.8; N, 7.4; H, 5.3; S, 8.5. FTIR (KBr, cm^−1^): 3434 (OH); 1664 (CN); 1638 (CN); 1339 (SO_2_); 1162 (SO_2_). ^1^H NMR (DMSO, δ_ppm_): 7.45 (t, *J* = 7.7 Hz, 2H); 7.25 (t, *J* = 7.4 Hz, 1H); 6.99 (d, *J* = 7.7 Hz, 2H); 3.73, 3.51 (2d, *J* = 14 Hz, 2H); 2.87 (d, *J* = 4.7 Hz, 1H); 2.33 –2.24 (m, 1H); 2.19–2.09 (m, 1H); 1.82–1.70 (m, 2H); 1.01 (s, 3H); 0.83 (s, 3H). ^13^C NMR (DMSO, δ_ppm_): 186.2 (C2); 167.9 (C3); 148.9 (C_ipso_); 129.4, 126.1, 120.2 (C_Ph_); 63.1 (C1); 51.1 (C4); 49.4 (C8); 46.4 (C7); 27.8, 23.5 (C5, C6); 19.3, 17.6 (C9, C10).

[Ag(OH)(SO_2_NC_10_H_13_NC_6_H_4_NH_2_−4)]·CH_3_COOH (**10f**)—A solution of SO_2_NC_10_H_13_N(C_6_H_4_)NH_2_ (7, 95 mg; 0.30 mmol) in ethanol (5 mL) was added to a solution of silver acetate (50 mg; 0.30 mmol) in water (5 mL) and the suspension was stirred overnight. A dark brilliant compound was obtained that was filtered and dried under vacuum. Yield 46%. Elem. Anal. for AgC_18_H_24_N_3_O_5_S, Exp.: C, 43.1; N, 8.4; H, 4.7; S, 6.5. Calc.: C, 43.0; N, 8.4; H, 4.8; S, 6.4. IR (KBr cm^−1^): 3437, 3366 (NH); 1743 (CO); 1643 (CN); 1564 (COO); 1335 (SO_2 sym_); 1160 (SO_2 asym_). ^1^H NMR (DMSO, δ_ppm_): 6.98 (d, *J* = 8.5 Hz, 2H); 6.64 (d, *J* = 8.5 Hz, 2H); 5.70 (s, 2H); 3.64, 3.41 (2d, *J* = 14 Hz, 2H); 3.15 (d, *J* = 3.9 Hz, 1H); 2.34−2.18 (m, 2H); 1.85 (s, 3H); 1.76−1.65 (m, 2H); 1.04 (s, 3H); 0.77 (s, 3H). ^13^C NMR (DMSO, δ_ppm_): 187.0 (C2); 159.4 (C3); 149.5 (C_ipso_); 136.0, 125.8, 113.8 (C_Ph_); 62.6 (C1); 51.6 (C4); 49.2 (C8); 47.0 (C7); 28.4, 23.0 (C5, C6); 19.0, 17.8 (C9, C10).

[Ag(OH)(SO_2_NC_10_H_13_NC_6_H_4_CH_3_−4)] (**11f**)—A solution of SO_2_NC_10_H_13_N(C_6_H_4_)CH_3_ (11, 95 mg; 0.30 mmol) in ethanol (5 mL) was added to a solution of silver acetate (50 mg; 0.30 mmol) in water (5 mL) and the suspension was stirred overnight. A yellow compound was obtained by filtration. Yield 63%. Elem. Anal. for AgC_17_H_21_N_2_O_3_S, Exp.: C, 46.4; N, 6.5; H, 4.9; S, 7.3. Calc.: C, 46.3; N, 6.4; H, 4.8; S, 7.3. IR (KBr cm^−1^): 1661 (CN); 1633 (CN); 1336 (SO_2 sym_); 1160 (SO_2 asym_). ^1^H NMR (CD_3_CN, δ_ppm_): 7.26 (d, *J* = 8.1 Hz, 2H); 6.92 (d, *J* = 8.2 Hz, 2H); 3.49, 3.27 (2d, *J* = 14 Hz, 2H); 3.00 (d, *J* = 4.8 Hz, 1H); 2.36 (s, 3H); 2.34–2.16 (m, 2H); 1.90–1.74 (m, 2H); 1.06 (s, 3H); 0.86 (s, 3H). ^13^C NMR (CD_3_CN, δ_ppm_): 187.6 (C2); 168.2 (C3); 147.5, 137.4 (C_ipso_); 130.7, 121.8 (C_Ph_); 64.2 (C1); 52.5 (C4); 50.6 (C8); 47.5 (C7); 29.2, 24.5 (C5, C6), 21.0 (CH3), 20.0, 18.2 (C9, C10).

### 3.3. Cyclic Voltammetry Studies

The redox properties of the complexes and ligands were studied by cyclic voltammetry using a three compartments cell equipped with a Pt wire electrode and interfaced with VoltaLab PST050 equipment. The cyclic voltammograms were obtained from NBu_4_BF_4_ solutions in CH_3_CN (0.10 M) used as electrolyte. At least ten cyclic voltammograms of each compound was obtained. The double electrode double layer was renewed between each cycle by bubbling nitrogen in the cell. The potentials were measured in volts (±10 mV) versus SCE at 200 mV/s using [Fe(η^5^-C_5_H_5_)2]^0/+^ (=0.382 V; CH_3_CN) as internal reference.

### 3.4. X-ray Diffraction Analysis

X-ray data for [Ag(NO_3_)(NC_10_H_14_NC_6_H_4_)] (6a) and SO_2_NC_10_H_14_NC_6_H_4_CH_3_−4 (11) were collected using a Bruker AXS-KAPPA APEX II area detector apparatus equipped with a graphite-monochromated Mo Kα (λ = 0.71073 Å) and was corrected for Lorentz polarization and, empirically, for absorption effects. The structures were solved by direct methods using SHELX97 [27] and refined by full matrix least squares against F^2^ using SHELX97 all included in the suite of programs WinGX v2020.1 for Windows [28]. The non-hydrogen atoms were refined anisotropically and the H atoms were inserted in idealized positions and allowed to refine riding on the parent atom. Crystal data and refinement parameters are summarized in Table 2. Illustrations of the molecular structures were made with ORTable 3. [28].

The Cambridge Crystallographic Data Centre (CCDC 2044258∓2044259) contains the supplementary crystallographic data for this article. The X-ray data can be obtained free of charge via www.ccdc.cam.ac.uk/conts/retrieving.html (or from the CCDC, 12, Union Road, Cambridge CB2 1EZ, UK; fax: +44 1223 336033 or deposit@ccdc.cam.ac.uk).

### 3.5. DFT Calculations

DFT calculations were carried out using GAMESS-US [29] version R3 with a CAMB3LYP function [30], with 65% Hartree-Fock (HF) exact exchange at long range and 19% at short range, using an SBKJC basis set. An incremental approach was used to build the oligomer chains from a single “mer” unit, by adding successive new metal centers to the seed unit. The dimeric oligomer was used to probe the conformational space of possible isomers. Once a growing trend was established (from the dimer), the new metal units were added without further probing of isomers. The four-camphor oligomer chain was included in the paper as an illustrative example. The optimized structures were confirmed as minimums by Hessians with positive eigenvalues and six near zero frequencies.

### 3.6. Bacterial Strains and Minimum Inhibitory Concentration Assays

The bacterial strains *Staphylococcus aureus* Newman, *Pseudomonas aeruginosa* 477, *Escherichia coli* ATCC25922, and *Burkholderia contaminans* IST408 were used in the present work and kept as frozen stock suspensions at –80 °C. When in use, bacterial cultures were maintained in Lennox Broth solid medium (Sigma-Aldrich, St. Louis, USA). Minimum inhibitory concentration (MIC) of the complexes under study were assessed by microdilution assays using Muller Hinton broth (MH) (Sigma-Aldrich, St. Louis, USA) as previously described [11,12,18,19]. Bacterial growth was assessed by measuring the cultures optical density at 640 nm (OD_640_). The MIC values were estimated by fitting the OD_640_ mean values, resulting from at least three independent experiments carried out in duplicate, with a Gompertz modified equation [12,13,14].

## 4. Conclusions

The antibacterial activities of the Ag(I) camphor imine (Series **a**, **b**, and **c**) and camphor sulfonimine imine (Series **d**, **e**, and **f**) complexes were evaluated against *E. coli*, *P. aeruginosa*, *B. contaminans*, and *S. aureus* through determination of MIC values. The analysis and rationalization of the results show that the imine or sulfonimine character of the camphor ligands considerably modify the behavior of the complexes in what concerns their sensitivity to the metal core and substituents on the camphor ligand. The Ag(I) camphor imine complexes are highly sensitive to nitrate or hydroxide co-ligands, showing that the {Ag(OH)} complexes are bacteriologically more active than the {Ag(NO_3_)} complexes. No such effect was detected on the camphor sulfonimine Ag(I) complexes. Additionally, the amine substituent (Y = NH_2_) at the {Ag(NO_3_)} camphor imine complexes (Series **a**) considerably enhances the antibacterial activity, while no such effect is observed for the camphor sulfonimine silver nitrate complexes (Series **d**). 

The redox properties of the {Ag(OH)} and {Ag(NO_3_)} complexes are considerably different; Ag(OH) complexes reduce at much lower potentials than {Ag(NO_3_)} (except ([Ag(OH)(OC_10_H_14_NOH)], **2c**), irrespective of the camphor imine or sulfonimine ligand. Silver hydroxy complexes resist better to Ag(I) to Ag^0^ reduction, and this possibly contributes to the enhancement of the complexes’ antimicrobial activity.

Overall, the herein results show that it is possible to tune the antibacterial activity of the camphor silver complexes through replacement of the co-ligand (NO_3_^−^ by OH^−^) or re-design of the camphor ligand; camphor sulfonimine complexes tend to be more active than the related camphor imine ones. Additionally, it was also possible to verify that a relationship exists between the redox properties of the complexes and their antibacterial activity. Results herein are based on the synthesis and characterization of nine new Ag(I) complexes, the analytical and spectroscopic characteristics of which are displayed in Section 3. 

## Figures and Tables

**Figure 1 antibiotics-10-00135-f001:**
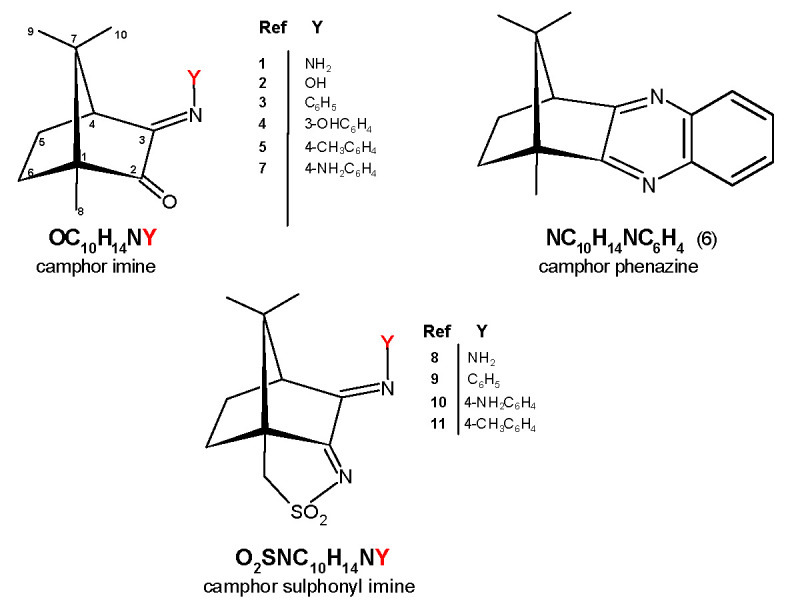
Ligands used in this work: camphor imine (**1**−**5** and **7**, showing numbering scheme), camphor phenazine (**6**), and camphor sulfonimine (**8**−**11**).

**Figure 2 antibiotics-10-00135-f002:**
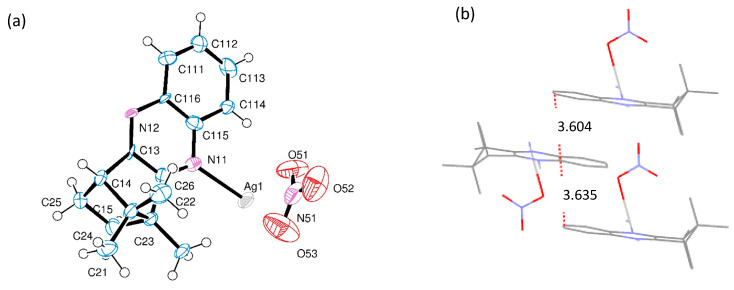
(**a**) ORTEP drawing for [Ag(NC_10_H_14_NC_6_H_4_)]NO_3_ (**6a**) showing labelling scheme. (**b**) Sequential layers showing distances between parallel phenyl rings. (**c**) Polymeric arrangement of **6a**.

**Figure 3 antibiotics-10-00135-f003:**
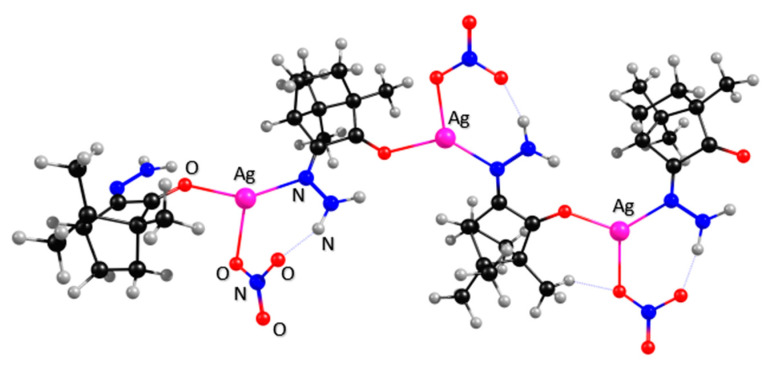
Calculated structural arrangement of [Ag(NO_3_)(OC_10_H_14_NNH_2_)] (**1a**) showing hydrogen bonding between the camphor imine substituent (Y = NH_2_) and the nitrate (NO_3_^−^) co-ligand.

**Figure 4 antibiotics-10-00135-f004:**
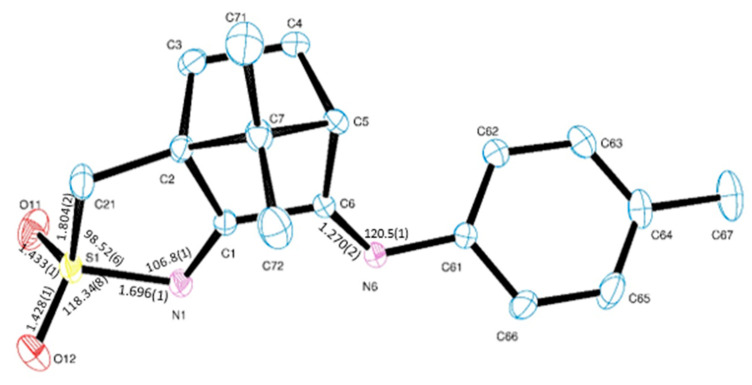
ORTEP drawing for SO_2_NC_10_H_13_NC_6_H_4_CH_3_−4 (**11**) showing labelling scheme and selected bond lengths (angstrom) and angles (deg).

**Figure 5 antibiotics-10-00135-f005:**
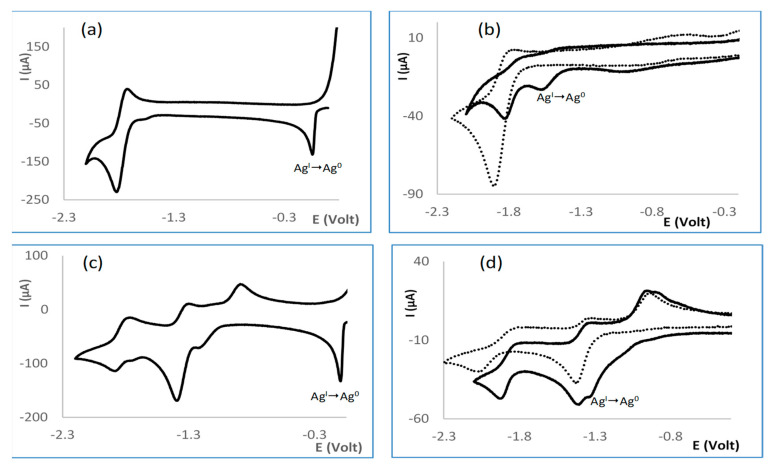
Partial cyclic voltammograms for (**a**) [Ag(NO_3_)(OC_10_H_14_NC_6_H_4_CH_3_)_2_] (**5a**); (**b**) [Ag(OH)(OC_10_H_14_NC_6_H_4_NH_2_)] (**7c**); (**c**) [Ag(NO_3_)(SO_2_NC_10_H_13_NC_6_H_5_)_2_] (**9d**); (**d**) [Ag(OH)(SO_2_NC_10_H_13_NC_6_H_4_NH_2_)_2_] (**10f**). Dashed lines at (**b**,**d**) refer to free ligands.

**Figure 6 antibiotics-10-00135-f006:**
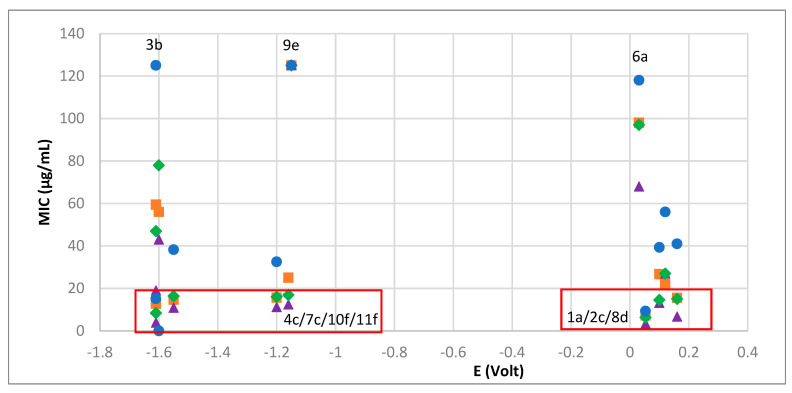
MIC values versus Ag(I)→Ag(0) reduction potentials for: *S. aureus* (blue circles), *E. coli* (orange squares), *B. contaminans* (green losangles), *P. aeruginosa* (purple triangles).

**Table 1 antibiotics-10-00135-t001:** Minimum inhibitory concentration (MIC) values and Ag^I^→Ag^0^ reduction potential for complexes.

.		E (Volt) ^a^	MIC (μg/mL)			
			Gram-Negative			Gram-Positive
COMPLEX	Y	Ag^I^→Ag^0^	*E. coli* ATCC25922	*P. aeruginosa* 477	*B. contaminans* IST408	*S. aureus* Newman
[Ag(NO_3_)(OC_10_H_14_NY)]	NH_2_ **1a**	0.12	22 ± 1	27 ± 1	27 ± 1	56 ± 3
[Ag(NC_10_H_14_NY)]NO_3_ ^b^	C_6_H_4_ **6a**	0.031	98 ± 1	68 ± 1	97 ± 1	118 ± 2
[{Ag(OC_10_H_14_NY)}_2_(μ-O)] ^b^	C_6_H_5_ **3b**	−1.61 −2.09	59.4 ± 0.3	19.0 ± 3	47.0 ± 7	125 ± 0
[{Ag(OC_10_H_14_NY)}_2_(μ-O)] ^b^	C_6_H_4_CH_3_ **5b**	−1.60	56.0 ± 5.0	43.0 ± 11.0	78.0 ± 2.0	58 ± 2
[Ag(OH)(OC_10_H_14_NY)]	OH **2c**	0.053	7.2 ± 0.1	3.4 ± 0.1	6.4 ± 0.1	9.3 ± 1.1
[Ag(OH)(OC_10_H_14_NY)]	3-OHC_6_H_4_ **4c**	−1.61	12.8 ± 1.2	3.9 ± 0.1	8.4 ± 0.1	15.1 ± 4.6
[Ag(OH)(OC_10_H_14_NY)]	C_6_H_4_NH_2_ **7c**	−1.55 −1.80	14.6 ± 4.1	10.9 ± 3.3	16.4 ± 2.9	38.2 ± 2.6
[Ag(NO_3_)(SO_2_NC_10_H_13_NY)_2_] ^c^	NH_2_ **8d**	0.10 −1.52	26.7 ± 0.1	13.2 ± 0.2	14.6 ± 1.1	39.3 ± 3.9
[Ag(NO_3_)(SO_2_NC_10_H_13_NY)_2_]	C_6_H_5_ **9d**	0.16 −1.13 ^d^ −1.62	15.5 ± 1.9	6.7 ± 0.3	15.1 ± 1.9	41 ± 6.3
[{Ag(SO_2_NC_10_H_13_NY)_2_}_2_(μ-O)]	C_6_H_5_ **9e**	−1.15 ^d^	125 ± 0	125 ± 0	>125	125 ± 0
[Ag(OH)(SO_2_NC_10_H_13_NY)]	C_6_H_4_NH_2_ **10f**	−1.20 −1.29 −1.74	15.6 ± 5.0	11.2 ± 2.6	16.1 ± 2.4	32.5 ± 1.7
[Ag(OH)(SO_2_NC_10_H_13_NY)]	C_6_H_4_CH_3_ **11f**	−1.16 ^d^ −1.66	25 ± 2	12.4 ± 1.1	16.9 ± 1.3	41 ± 2
Ag(CH_3_COO)_2_		−0.043 −1.33	30.9 ± 0.4	16 ± 3	12 ± 2	29.5 ± 0.1
AgNO_3_ ^c^		0.18	47	39	74	73

^a^ In Bu_4_NBF_4_/CH_3_CN (0.1 M) using a Pt wire working electrode. Potentials in volt (±10 mV) measured versus Saturated Calomel Electrode (SCE) using Fe(C_5_H_5_)_2_]^0/+^ (*E* = 0.38 V) as internal reference. ^b^ Data from [14]. ^c^ Data taken from reference [13]. ^d^ Quasi-reversible wave $E_{1/2}^{red}.$

**Table 2 antibiotics-10-00135-t002:** Crystallographic data for [Ag(NO_3_)(NC_10_H_14_NC_6_H_4_)] (**6a**) and SO_2_NC_10_H_14_NC_6_H_4_CH_3_ (**11**).

	[Ag(NO_3_)(NC_10_H_14_NC_6_H_4_)]	SO_2_NC_10_H_14_NC_6_H_4_CH_3_-4
Empirical formula	Ag_2_N_6_C_32_H_36_O_6_	N_4_C_34_H_40_O_4_S_2_
Formula weight	816.42	632.82
Crystal system	Monclinic	Orthorhombic
Space group	C2	P2_1_2_1_2_1_
Unit cell dimensions		
a/Å	28.720 (5)	8.9160 (3)
b/Å	7.3319 (9)	12.0253 (4)
c/Å	7.2379 (9)	14.8492 (4)
α/deg	90	90
β/deg	95.29 (1)	90
γ/deg	90	90
Volume (Å^−3^)	1517.6 (4)	1592.10 (9)
Z, Dcal (g/cm^3^)	2, 1.787	2, 1.320
Absorption coefficient (mm^−1^)	1.347	0.212
F(000)	824	672
Crystal size (mm^3^)	0.3 × 0.2 × 0.3	0.2 × 0.3 × 0.2
θ range for data collection (deg)	1.4 to 33.0	2.2 to 32.6
Index ranges	−37 ≤ h ≤ 43, −8 ≤ k ≤ 11, −11 ≤ l ≤ 11	−13 ≤ h ≤ 13, −16 ≤ k ≤ 18, −22 ≤ l ≤ 22
Reflections collected / unique	7272/4840 [R(int) = 0.048]	22635/5803 [R(int) = 0.050]
Data/restraints/parameters	4840/1/211	5803/0/202
Final R (observed)	R1 = 0.065, wR2 = 0.19	R1 = 0.034, wR2 = 0.092

## Data Availability

The X-ray data can be obtained free of charge via www.ccdc.cam.ac.uk/conts/retrieving.html (or from the CCDC, 12, Union Road, Cambridge CB2 1EZ, UK; fax: +44 1223 336033 or deposit@ccdc.cam.ac.uk).

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
