# Peer review of "Key Parameters on the Antibacterial Activity of Silver Camphor Complexes"

_antibiotics, 2021, doi:10.3390/antibiotics10020135_

Round 1
Reviewer 1 Report
In the manuscript entitled “Key parameters on the antibacterial activity of silver camphor
complexes” by Costa et al. (Ms. ID:antibiotics-1090476) nine new complexes with camphor imine or camphor sulfonimine ligands were prepared and characterized and their antibacterial activities were estimated through the determination of the MIC values. The authors related the MIC values obtained with different characteristics of the complexes studied. Besides, the redox properties of the complexes were determined using cyclic voltammetry. The work is well written and the experimental results seemed sound. However, there are certain issues that the authors need to clear up before publication in Antibiotics:
1.- Why the authors have used only one type of gram-positive bacteria? In order to compare the MIC data obtained for gram-negative and gram-positive strains more measurements using additional gram-positive bacteria will be needed.
2.- In the Materials and Methods section no description about the cyclic voltammetry measurements can be found. The authors should include a subsection describing how they have done the measurements, the number of repetitions, etc.
3.- Have the authors tried to find a correlation between the redox properties of the complexes and their MIC values? A plot of the redox potentials vs. MIC for example.
Author Response
Dear Reviewer,
We highly appreciated your comments and criticisms, thank you. Below find a point by point response to your criticisms:
Reviewer #1
In the manuscript entitled “Key parameters on the antibacterial activity of silver camphor complexes” by Costa et al. (Ms. ID:antibiotics-1090476) nine new complexes with camphor imine or camphor sulfonimine ligands were prepared and characterized and their antibacterial activities were estimated through the determination of the MIC values. The authors related the MIC values obtained with different characteristics of the complexes studied. Besides, the redox properties of the complexes were determined using cyclic voltammetry. The work is well written and the experimental results seemed sound. However, there are certain issues that the authors need to clear up before publication in Antibiotics:
1.- Why the authors have used only one type of gram-positive bacteria? In order to compare the MIC data obtained for gram-negative and gram-positive strains more measurements using additional gram-positive bacteria will be needed.
Answer: Thank you for the positive comments on the manuscript. The present manuscript results from research work that has been carried out in this topic and is built upon previous results, representing an advance in our efforts in the understanding of structure/antimicrobial properties of silver and camphor derivatives, envisaging the optimization of a structure with enhanced antimicrobial properties. Since all previous work, mentioned throughout the manuscript, was performed with S. aureus as a representative of pathogenic Gram-positive bacteria, the inclusion of additional Gram-positive bacteria in this work would require a huge amount of additional experimental work, which we are in no conditions to perform at this moment. In addition, comparisons would be limited, as previous work only included S. aureus as a Gram-positive representative. In our view, the main conclusions drawn would not substantially change. Nevertheless, we were careful in limiting conclusions to the Gram-positive S. aureus, and hypothesize that some similar results might be extended to other Gram-positive bacteria. Nevertheless, we appreciate the comment and in future work we will consider the inclusion of additional Gram-positive bacteria.
2.- In the Materials and Methods section no description about the cyclic voltammetry measurements can be found. The authors should include a subsection describing how they have done the measurements, the number of repetitions, etc.
Answer: Thank you for your comment. As requested, a subsection was included describing the experimental details on voltammetry measurements. Accordingly, the sentences related to voltammetry in section 4.1 were deleted.
See new lines 430-438, which now reads as follows:
“4.3 Cyclic Voltammetry studies
The redox properties of the complexes and ligands were studied by cyclic voltammetry using a three compartments cell equipped with a Pt wire electrode and interfaced with a VoltaLab PST050 equipment. The cyclic voltammograms were obtained from NBu4BF4 solutions in CH3CN (0.10 M) used as electrolyte. At least ten cyclic voltammograms of each compound was obtained. The double electrode double layer was renewed between each cycle by bubbling nitroge in the cell. The potentials were measured in Volts (± 10 mV) versus SCE at 200 mV/s using [Fe(η5-C5H5)2]0/+ (=0.382 V; CH3CN) as internal reference.”
3.- Have the authors tried to find a correlation between the redox properties of the complexes and their MIC values? A plot of the redox potentials vs. MIC for example.
Answer: We thank the reviewer for the suggestion. A plot of the redox potentials vs. MIC values was included, the new Figure 6 (see new page 11).
Reviewer 2 Report
General comments
The authors describe synthesis and characterization of nine new Ag(I) camphor complexes. The new compounds were tested for their antibacterial potency by determining their minimal inhibitory concentration (MIC) values against three Gram-negative and one Gram-positive bacterium. The chemical synthesis / characterization of the compounds is very detailed and well-written. It is a bit a pity that the set of tested bacteria is limited, especially concerning Gram-positives. Nevertheless, the study is of great interest and relevance. Finally, I understand that the following might be premature but maybe some note on expected toxicity / safety of these compounds in the host (animal/human) could be added, in case earlier studies with similar compounds have been conducted in the past.
On the formal side, the text was not carefully proofread and contains various typos / wrong cross references etc. which must be corrected.
Detailed comments
Lines 36 ff: please italicize all bacterial genus and species names
Line 39: better specify “third generation β-lactam antibiotics” and “as well as carbapenems” since these do not belong to the 3rd gen β-lactams
Line 57: provide an example for mentioned catalytic processes
Line 77: missing dot.
Line 80: the mention of “six sets” is a bit confusing since Fig. 1 looks like three sets, maybe this can be improved.
Line 115: introduce DFT abbreviation before first use
Line 130 / 174: insert cross reference.
Line 171: put “0” as superscript
Table 1: by which criterion were the thick lines placed? Maybe better separate compounds of the different families.
Line 231: which table?
Line 280: delete this
Line 386: for 4 hours I suppose.
Line 458: I suppose this should be a plus sign fax: þ44 1223 336033
Line 486: better “determination” than “calculation”
Author Response
Dear Reviewer,
We highly appreciated your positive comments and criticisms, thank you. Below follows a point-by-point response to the comments and criticisms:
Reviewer #2
The authors describe synthesis and characterization of nine new Ag(I) camphor complexes. The new compounds were tested for their antibacterial potency by determining their minimal inhibitory concentration (MIC) values against three Gram-negative and one Gram-positive bacterium. The
chemical synthesis / characterization of the compounds is very detailed and well-written. It is a bit a pity that the set of tested bacteria is limited, especially concerning Gram-positives. Nevertheless, the study
is of great interest and relevance. Finally, I understand that the following might be premature but maybe some note on expected toxicity / safety of these compounds in the host (animal/human) could be added, in case earlier studies with similar compounds have been conducted in the past.
On the formal side, the text was not carefully proofread and contains various typos / wrong cross references etc. which must be corrected.
Answer: We appreciated the positive comments of the reviewer on our manuscript, and thank for the suggestions. As we stated in answers to Reviewer #1, the present manuscript results from research work that has been carried out in this topic and is built upon previous results, representing an advance in our efforts in the understanding of structure/antimicrobial properties of silver and camphor derivatives, envisaging the optimization of a structure with enhanced antimicrobial properties. Since all previous work, mentioned throughout the manuscript, was performed with S. aureus as a representative of pathogenic Gram-positive bacteria, the inclusion of additional Gram-positive bacteria in this work would require a huge amount of additional experimental work, which we are in no conditions to perform at this moment. In addition, comparisons would be limited, as previous work only included S. aureus as a Gram-positive representative. In our view, the main conclusions drawn would not substantially change. Nevertheless, we were careful in limiting conclusions to the Gram-positive S. aureus, and hypothesize that some similar results might be extended to other Gram-positive bacteria. Nevertheless, we appreciate the comment and in future work we will consider the inclusion of additional Gram-positive bacteria.
We also appreciate the suggestion to include in future studies the assessment of toxicity/safety of compounds using animal models and human cell lines. This is indeed a goal of our future work to advance investigation on the use of silver camphor complexes as therapeutic antimicrobial compounds. We apologize for the typos and wrong cross-references. A thorough revision was performed in the preparation of the revised version of the manuscript.
Detailed comments
Lines 36 ff: please italicize all bacterial genus and species names
Answer: Thanks. The correction was done.
Line 39: better specify “third generation β-lactam antibiotics” and “as
well as carbapenems” since these do not belong to the 3rd gen β-lactams
Answer: Thanks for the correction. As suggested, the new line 39 now reads as follows: “some strains are resistant to the third generation β-lactam antibiotics as well as to carbapenems”
Line 57: provide an example for mentioned catalytic processes
Answer: Thank you for the suggestion. The sentence was rephrased: “The electronic and steric characteristics of coordination compounds may also switch catalytic redox processes (on/off), reducing bacteria growth, e.g. through ROS generation [4].” Please see new lines 57-58.
Line 77: missing dot.
Answer: The dot was added.
Line 80: the mention of “six sets” is a bit confusing since Fig. 1 looks like three sets, maybe this can be improved.
Answer: Thanks for the comment. The sentence was rephrased to: “Six sets of silver complexes were synthesized based on camphor-type ligands (Figure 1) aiming at enhancing the antibacterial …”. See new line 80.
Line 115: introduce DFT abbreviation before first use
Answer: Thanks for the comment. The abbreviation was introduced and it now reads as follows “1a and 6a, calculations using Density Functional Theory (DFT)…”. See new line 115.
Line 130 / 174: insert cross reference.
Answer: Thanks for the suggestion. However, we have already inserted a reference in line 130, which reads as follows: “or even linear geometries [13].” Reference in line 175 was also corrected to “[19]”.
Line 171: put “0” as superscript
Answer: We apologize for this format error. Corrected. See new line 172: “Table 1. MIC values and AgI"Ag0 reduction potential for complexes”
Table 1: by which criterion were the thick lines placed? Maybe better separate compounds of the different families.
Answer: Thanks for comment and suggestion. As suggested, we have placed the thick lines according to their different families. See new Table 1.
Line 231: which table?
Answer: We apologize for the missing info. It is Table 1. See correction in new line 231
Line 280: delete this
Answer: We apologize. Done
Line 386: for 4 hours I suppose.
Answer: Thanks for the observation. It is indeed four hours. Corrected.
Line 458: I suppose this should be a plus sign fax: þ44 1223 336033
Answer: Thanks for the observation. It is indeed a plus sign. Corrected to : +44 1223 336033
Line 486: better “determination” than “calculation”
Answer: Thanks for the remark. We have made the indicated change.
Round 2
Reviewer 1 Report
The revised version of the manuscript is suitable for publication in Antibiotics.